# Atypical calcium regulation of the PKD2-L1 polycystin ion channel

**Paul G DeCaen[1,2†], Xiaowen Liu[1,2†], Sunday Abiria[1,2], David E Clapham[1,2]***

[1]Department of Cardiology, Howard Hughes Medical Institute, Boston Children's Hospital, Boston, United States; [2]Department of Neurobiology, Harvard Medical School, Boston, United States

**Abstract** Native PKD2-L1 channel subunits are present in primary cilia and other restricted cellular spaces. Here we investigate the mechanism for the channel's unusual regulation by external calcium, and rationalize this behavior to its specialized function. We report that the human PKD2-L1 selectivity filter is partially selective to calcium ions ($Ca^{2+}$) moving into the cell, but blocked by high internal $Ca^{2+}$ concentrations, a unique feature of this transient receptor potential (TRP) channel family member. Surprisingly, we find that the C-terminal EF-hands and coiled-coil domains do not contribute to PKD2-L1 $Ca^{2+}$-induced potentiation and inactivation. We propose a model in which prolonged channel activity results in calcium accumulation, triggering outward-moving $Ca^{2+}$ ions to block PKD2-L1 in a high-affinity interaction with the innermost acidic residue (D523) of the selectivity filter and subsequent long-term channel inactivation. This response rectifies $Ca^{2+}$ flow, enabling $Ca^{2+}$ to enter but not leave small compartments such as the cilium.

## Introduction

**\*For correspondence:** dclapham@ enders.tch.harvard.edu

[†]These authors contributed equally to this work

Polycystic kidney disease proteins (PKDs), or polycystins (PC), are divided into two distinct gene families. The four PKD1 members (PKD1, PKD1-L1, PKD1-L2, PKD1-L3) are large proteins (~1700–4300 amino acids) with 11 putative transmembrane segments (TM) and a large autocleaved N-terminal extracellular domain. In contrast, the three PKD2 members are often included in the TRP ion channel family because they have 6 TM domains and a putative selectivity filter loop between TM5 and TM6 (*Ramsey et al., 2006*). These include PKD2, (PC2, TRPP1; formerly TRPP2), PKD2-L1 (PC2-L1, TRPP2; formerly TRPP3) and PKD2-L2 (PC2-L2, TRPP3, formerly TRPP5) (*Wu et al., 2010*). Members of the PKD1 and PKD2-subfamilies are often reported to associate in the plasma membrane, although the nature of these complexes is not understood. PKD1 + PKD2 were purported to form mechanosensitive ion channels in the primary cilia of kidney collecting duct epithelia (*Nauli and Zhou, 2004*; *Nauli et al., 2003*), a hypothesis that has recently been challenged (*DeCaen et al., 2013*; *Delling et al., 2016*). In humans, autosomal dominant polycystic kidney disease (ADPKD) is associated with loss-of-function mutations in genes that encode for either PKD1 or PKD2 (*Wu and Somlo, 2000*). Complete loss of either *Pkd1* or *Pkd2* in mice results in embryonic lethality with defects in formation of the kidney, pancreas and heart (*Boulter et al., 2001*; *Lu et al., 1997*; *Kim et al., 2000*; *Wu et al., 2000*). As in human ADPKD, adult *Pkd2^{WS25/−}* mice have kidney cysts, renal failure, and die early (*Wu et al., 2000*). PKD1-L1 + PKD2-L1 heteromers form a calcium channel complex in the primary cilia from embryonic fibroblasts and retinal pigmented epithelial cells (*DeCaen et al., 2013*). Mice lacking *Pkd2-l1* exhibit a form of heterotaxy (intestinal malrotation) in ~50% of offspring, suggesting that the channel modulates the ciliary Sonic Hedgehog (SHh) pathway during early development (*Delling et al., 2013*). A putative PKD1-L3 + PKD2-L1 channel was reported in mouse taste buds and proposed to be the acid receptor required for sour taste

**eLife digest** Most of our cells have a single tiny-hair like structure called a primary cilium that projects outwards from the cell surface. Many cilia contain an ion channel protein called PKD2-L1 that allows calcium ions to pass through the membrane that surrounds each cell.

There are many different calcium channels and they are found in a variety of locations in cells to control the levels of calcium ions within various cell compartments. Channels on the cell surface allow calcium ions from the external environment to enter a compartment called the cytoplasm. Under normal conditions, calcium ions always flow into cells because they are much more abundant outside the cell than inside. Despite the absence of a membrane barrier between cilia and cytoplasm, calcium ions are maintained at a higher level in the cilium than in the cytoplasm. How is this difference maintained?

DeCaen, Liu et al. now show that PKD2-L1 is first stimulated and then inactivated by calcium ions inside the cilia. Under controlled conditions, calcium ions exiting PKD2-L1 block the channel and trigger its long-term inactivation. This inactivation can be overcome by preventing calcium ions from accumulating inside the cell.

Further experiments show that cytoplasmic regions of the PKD2-L1 protein that are able to bind to calcium ions are not responsible for this unusual channel behavior. DeCaen, Liu et al. propose that the strange effect of calcium ions on the channel acts to maintain calcium ions in cilia at higher levels than those found in the cytoplasm. Future challenges include understanding how PKD2-L1 is stimulated by calcium ions and to find out the consequences of this activity.

(*Kawaguchi et al., 2010*), but *Pkd2-L1* knockout mice do not appear to have a deficit in sour taste perception (*Nelson et al., 2010*).

Of all the members of the PKD family, we found that only PKD2-L1 forms a functional homotetrameric channel when heterologously expressed on the plasma membrane (*DeCaen et al., 2013*). Information about the physiological function of the endogenous homomeric PKD2-L1 is sparse. Emerging evidence suggests that single channel openings of homomeric PKD2-L1 channels are sufficiently large (>100 pS) to generate action potentials in the medullo-spinal cerebrospinal fluid contacting neurons (CSF-cNs) located in the ependymal layer of the brainstem (*Orts-Del'Immagine et al., 2014*; *Orts-Del'immagine et al., 2012*). Since channels containing PKD2-L1 subunits are constitutively-active and calcium-permeant, dysregulation of their activity could lead to aberrant cytoplasmic calcium regulation, especially in smaller compartments such as the primary cilia of epithelial cells (*DeCaen et al., 2013*), the dendritic bulb of cerebrospinal fluid-contacting neurons (*Orts-Del'Immagine et al., 2014*) and the apical processes of the type III taste receptors (*Huang et al., 2006*; *Ishimaru et al., 2010*). Based on results using GCaMP3-targeted calcium sensors and current clamp recordings, primary cilia membranes were depolarized and had higher resting calcium concentrations ($\sim -20$ mV, $[Ca^{2+}]_{in}$ ~700 nM) compared to the cell body (-50 mV, $[Ca^{2+}]_{in}$ < 100 nM) (*Delling et al., 2013*). Interestingly, the endogenous PKD1-L1 + PKD2-L1 channel complex inactivates in response to high cytoplasmic $[Ca^{2+}]$ when measured directly from the primary cilium ($IC_{50}$ =540 nM) (*DeCaen et al., 2013*). When measured through heterologous co-expression, the PKD1-L3 + PKD2-L1 channel was initially activated, then inactivated by extracellular $Ca^{2+}$ (5–10 mM) (*Chen et al., 2015*), which was also observed when homomeric PKD2-L1 channels were expressed in in *Xenopus* oocytes (*Chen et al., 1999*). It is unclear if calcium regulates the channels composed of PKD2-L1 directly or indirectly through an internal or external binding site.

Here we have taken the reductionist approach of measuring the PKD2-L1 homomeric channel in a heterologous system in order to understand its mechanism of calcium-dependent inactivation. We first confirm previous studies that PKD2-L1 is modestly voltage-dependent (*Shimizu et al., 2009*) and show that this voltage dependence is independent of any divalent ion blocking mechanism. We then demonstrate that PKD2-L1 inactivation is initiated by the accumulation of internal $Ca^{2+}$. Surprisingly, we find that removal of the cytoplasmic EF-hands and coiled-coil motifs do not alter PKD2-L1 inactivation. We observe that $Ca^{2+}$ ions conduct inwardly through the selectivity filter of PKD2-L1, whereas outward $Ca^{2+}$ conductance blocks the channel, followed by long-term inactivation. This

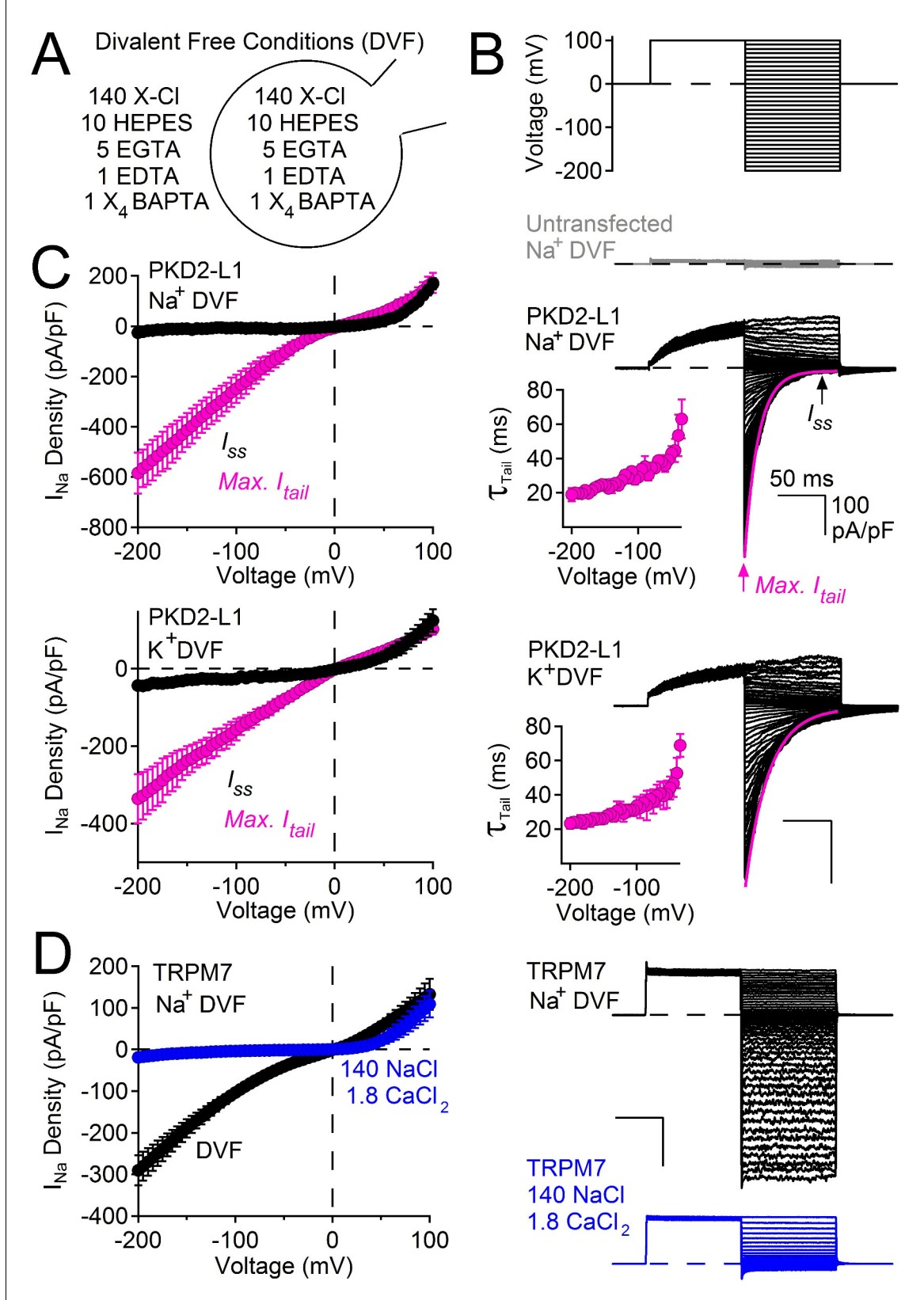

**Figure 1.** PKD2-L1 channel deactivation is voltage-dependent. (**A**) Diagram depicting symmetrical divalent free (DVF) conditions. (**B**) *Top*, voltage protocol used to generate tail currents. Representative currents from untransfected HEK cells (gray traces) and those expressing PKD2-L1 channels

*Figure 1 continued on next page*

*Figure 1 continued*

(black traces). Tail currents were fit to a single exponential (magenta trace), averaged and plotted relative to voltage (Inset, Error ± SEM, N = 4–7 cells). (C) Sodium current density-voltage relationships measured at the maximum of the tail current and at steady state. Results from repeated trials of the voltage protocol are indicated (Error ± SEM, N = 4–7 cells). (D) *Right*, Representative TRPM7 DVF Na$^+$ currents recorded under (black traces) and after the addition of 1.8 mM CaCl$_2$ (blue traces). *Left*, Resulting TRPM7 current density-voltage relationships (Error ± SEM, N = 8 cells).

The following figure supplement is available for figure 1:

**Figure supplement 1.** Decay of hyperpolarization-induced tail currents from PKD2-L1 channels.

type of Ca$^{2+}$-dependence appears to be unique to PKD2-L1 among TRP channels tested to date and is triggered by Ca$^{2+}$ coordination with D523 in the selectivity filter, which is also the site responsible for divalent metal and trivalent lanthanide metal block.

## Results

### PKD2-L1 channels are voltage-dependent, even in the absence of divalent ions

The voltage dependence of murine PKD2-L1 (previously named TRPP2, now called TRPP1) was previously characterized under conditions of high (11 mM) internal magnesium (*Shimizu et al., 2009*). To separate rectification due to divalent ion block (*Voets et al., 2003*; *Nadler et al., 2001*; *Lucas et al., 2003*; *Topala et al., 2007*) from intrinsic voltage dependence, we first examined human PKD2-L1 in symmetrical 144 mM [Na$^+$] in the absence of divalent ions (free Ca$^{2+}$ and Mg$^{2+}$ <1 nM, *Figure 1A*). From a holding potential of 0 mV, we applied a +100 mV prepulse and then measured tail currents after immediate hyperpolarization to varying potentials (*Figure 1B,C*). Plots of the instantaneous maximum tail currents (Max. I$_{tail}$) were nearly linear (ohmic) at all potentials. However, the steady-state current (I$_{ss}$) measured after the tail current decay outwardly rectified at potentials above +50 mV. At negative potentials PKD2-L1 appears to simply deactivate as estimated by the rate constants of tail current decay (τ = 20–60 ms, *Figure 1 B*). In the on-cell configuration, the decay of ensemble-averaged single channel events closely matched macroscopic tail current decay (*Figure 1—figure supplement 1*).

We considered the possibility of voltage or time-dependent monovalent ion block of the pore in the absence of divalent ions. However, single channel amplitudes appear to remain constant over the time course of the pulse, indicating a lack of fast (flicker) block (*Figure 1—figure supplement 1*). Also, replacing Na$^+$ by K$^+$ did not change the current amplitude or rate of decay (*Figure 1C*). Although we cannot rule out a site that binds and blocks all inward monovalent cations, we argue that the most likely interpretation is that the channel simply deactivates after hyperpolarization (see Discussion). It is important to note that the onset of the PKD2-L1 tail current is very fast (<120 μs, data not shown), without an obvious 'hook' to indicate overlapping activation and inactivation time courses as observed in delayed rectifying potassium channels (hERG and Kv3.1b) (*Labro et al., 2015*; *Vandenberg et al., 2012*). The tail currents of PKD2-L1 are unusual compared to other cation-nonselective TRP channels. For example, currents conducted by TRPM7 are ohmic and do not decay upon membrane hyperpolarization (*Figure 1D*). However, TRPM7 becomes outwardly rectifying when calcium is added to the external sodium solution (and, as is well-known for TRPM6/7, blocked by internal Mg$^{2+}$) (*Topala et al., 2007*; *Li et al., 2007*), indicating that TRPM7 rectifies in the presence of a physiological level of extracellular calcium, and possibly other divalent ions.

Taken together, these data demonstrate that human PKD2-L1 channels are voltage-dependent, as previously described for the murine orthologue by Nilius and colleagues (*Shimizu et al., 2009*). Thus, PKD2-L1 voltage dependence which is independent of cation block is likely shared between murine and human orthologs and may be a unique among the TRP family of ion channels. The following experiments focus on another unusual feature of PKD2-L1 we call 'long-term inactivation' which is dependent on the accumulation of internal calcium ions.

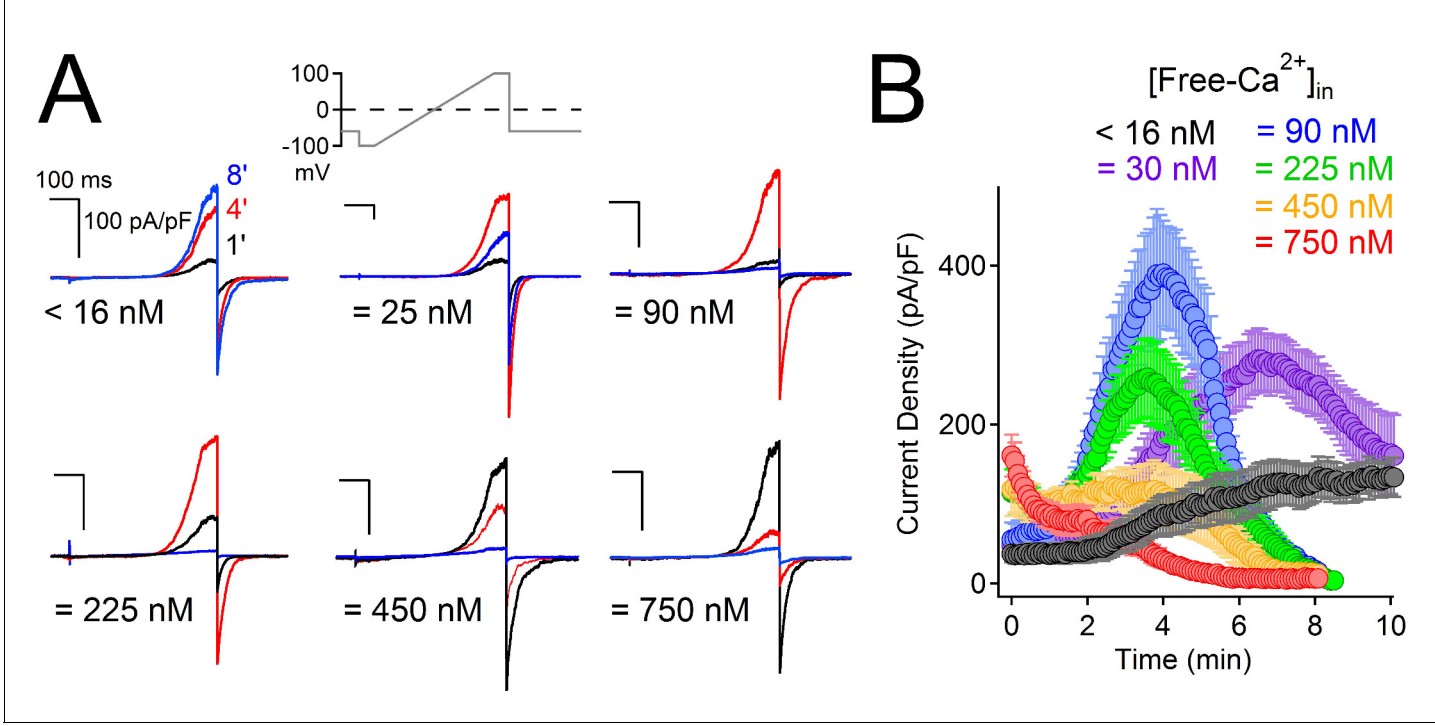

**Figure 2.** High intracellular $Ca^{2+}$ irreversibly inactivates PKD2-L1. (A) Representative PKD2-L1 currents captured at 1, 4 and 8 min time points recorded with the indicated buffered [free- $Ca^{2+}$]; 2 mM $Ca^{2+}$ in the external solution. (B) Time course of outward peak current density recorded under the indicated conditions (Error ± SEM, N = 7–10 cells).

The following figure supplement is available for figure 2:

**Figure supplement 1.** The time course of PKD2-L1 inactivation correlates with internal $Ca^{2+}$ accumulation.

## PKD2-L1 channels are blocked by $Ca^{2+}$ exiting the cell

High external $Ca^{2+}$ (10 mM) potentiates human (*Chen et al., 1999*) but not murine PKD2-L1 (*Shimizu et al., 2009*). To examine calcium's modulation of the current under physiological $Ca^{2+}$ conditions, we expressed human PKD2-L1 and measured PKD2-L1 currents under voltage clamp in 2 mM $[Ca^{2+}]_{ex}$. As with high external calcium, PKD2-L1 current was initially potentiated but, interestingly, was then completely and apparently irreversibly inactivated over 8 min (*Figure 2—figure supplement 1A,B*). During the voltage ramp, we observed a negative shift of $E_{rev}$ from 8 to 0 mV, reflecting a loss in the available calcium-permeable PKD2-L1 channels and/or a shift in the ion composition of the patched cells (*Figure 2—figure supplement 1C*). Similar shifts in equilibrium potentials due to accumulation of internal cations have been reported for cells overexpressing $P_2X_2$ channels (*Li et al., 2015*). Since potentiation (4 min) and inactivation (8 min) were slow, we hypothesized that internal accumulation of $Ca^{2+}$ could account for the delayed onset of both. We integrated the tail currents triggered by −60 mV repolarizations to estimate the number of calcium ions accumulating within the cell (*Figure 2—figure supplement 1A,C*). We found that $Ca^{2+}$ moving through PKD2-L1 saturates the internal calcium buffers (5 mM BAPTA and intrinsic) and accumulates to concentrations >10 µM (*Figure 2—figure supplement 1D*). The time course of BAPTA saturation by PKD2-L1-mediated increases in $Ca^{2+}$ occurs after 4 min, suggesting that internal calcium accumulation is responsible for channel inactivation. Consistent with these observations, we varied internal $[Ca^{2+}]$ and observed that cells patched with high initial $[Ca^{2+}]$ ($\geq$ 450 nM) inactivates 1–4 min sooner than cells patched with lower internal free-$Ca^{2+}$ ($\leq$ 100 nM; *Figure 2A,B*). Finally, when the buffering strength is increased (15 mM BAPTA, 5 mM EGTA), PKD2-L1 currents potentiate, but do not inactivate, over a 10 min time course (black trace, *Figure 2B*).

To determine whether internal $Ca^{2+}$ could alter PKD2-L1 channel activity when the endogenous buffering conditions were undisturbed, we patch-clamped cells expressing PKD2-L1 in the on-cell

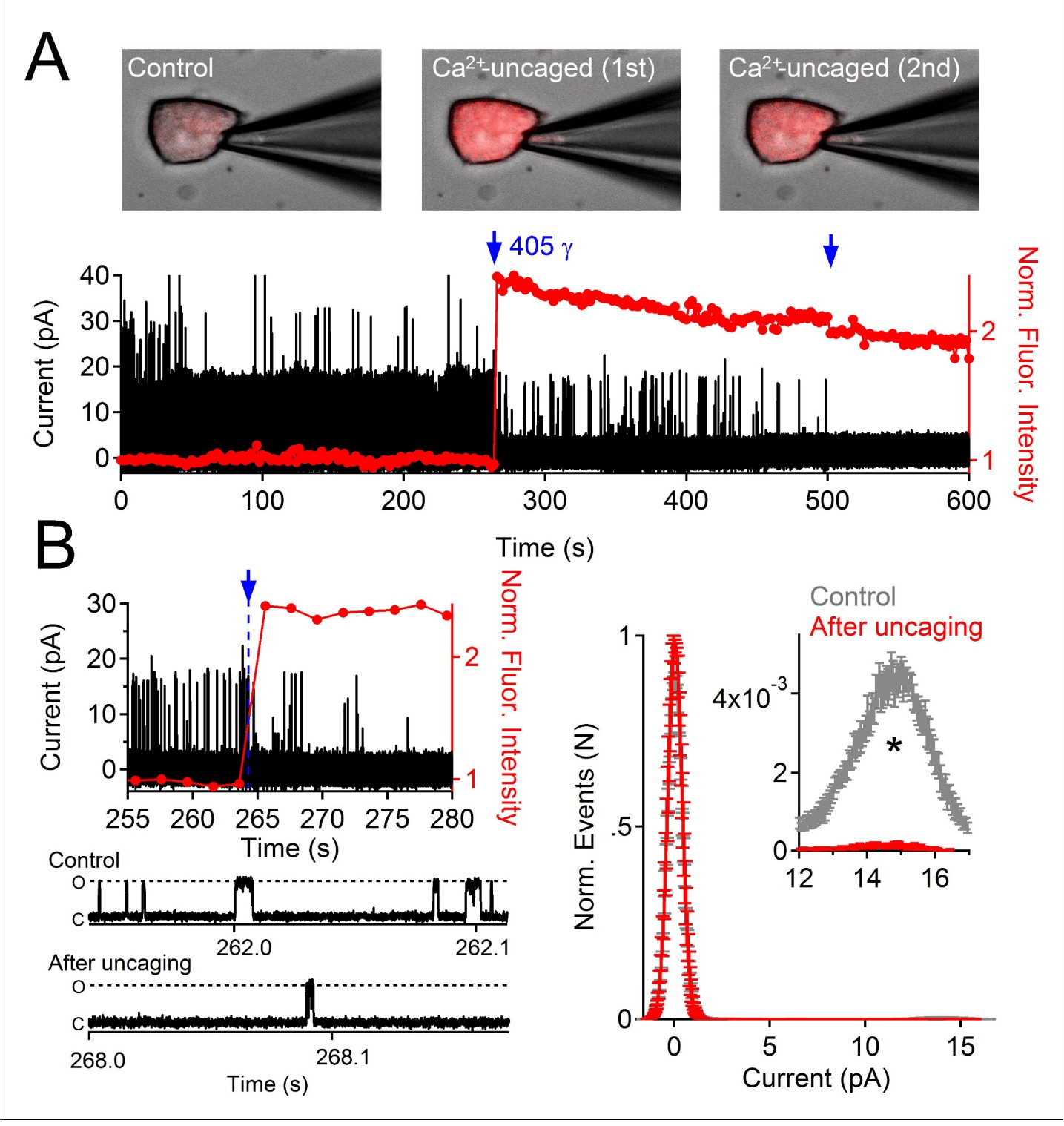

**Figure 3.** Uncaging internal $Ca^{2+}$ blocks outward single channel openings. (**A**) Outward PKD2-L1 single channel events measured in the on-cell configuration; holding potential = 80 mV. $Ca^{2+}$ dependent fluorescence was measured using Fluo-3; internal $Ca^{2+}$ was caged with NP-EGTA. Blue arrows indicate the time points at which $Ca^{2+}$ was uncaged using a 1 s 405 nm UV pulse. Images before and after uncaging are shown above the single channel record. (**B**) Expanded time scales of the record in A, illustrating the rapidity of current block. *Right*, Normalized open probability histograms measured in control (gray) and after uncaging cytosolic $Ca^{2+}$ (red). The open probability ($P_o$) of PKD2-L1 was significantly reduced after the UV pulse (N= 4 cells, Error ± SEM; asterisk indicates p<0.005; Prior to UV pulse, $P_o$ = 0.0057 ± 0.001; after UV pulse $P_o$ = 0.0002 ± 0.0002).

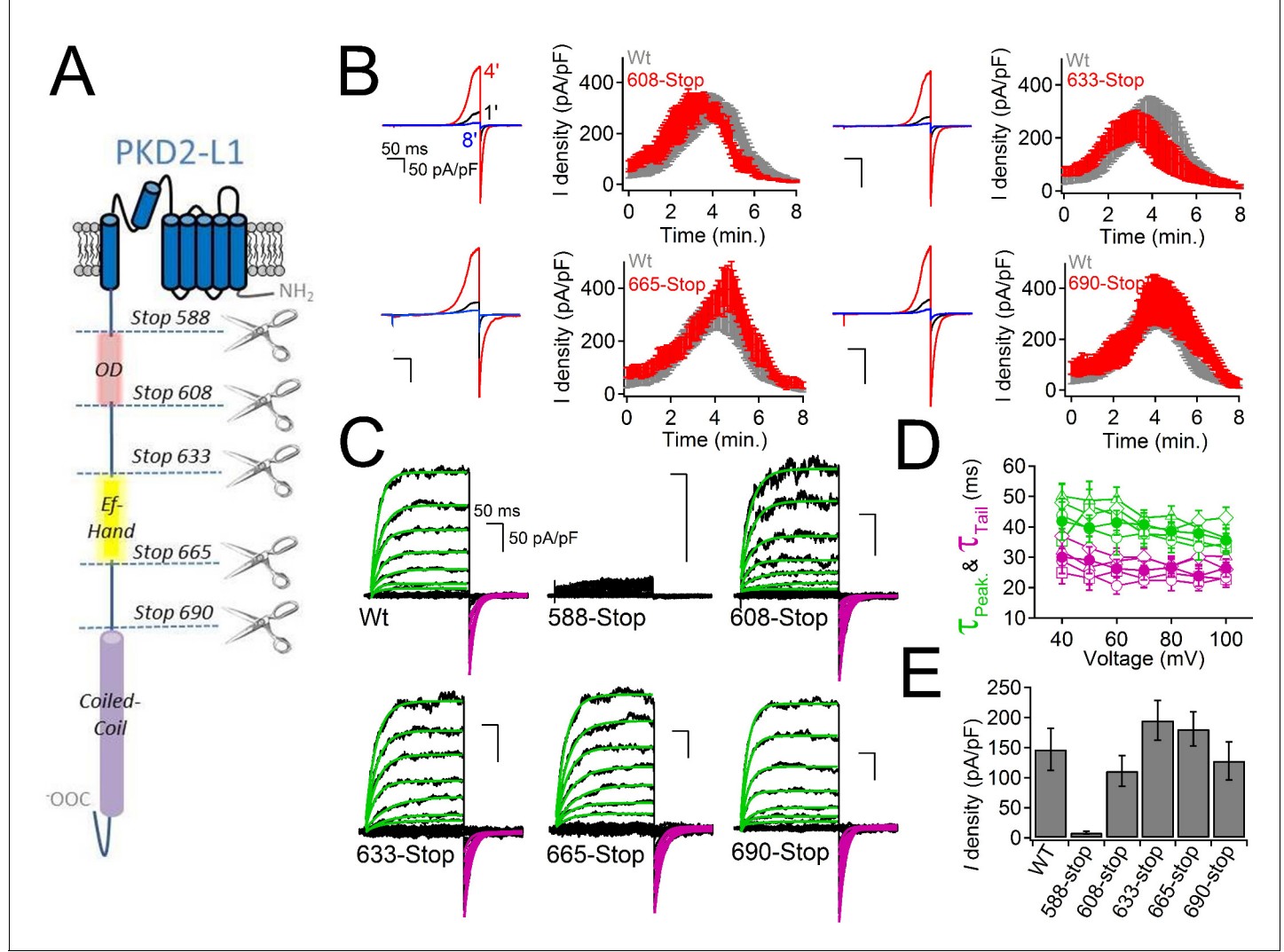

**Figure 4.** PKD2-L1 channel C-terminal truncations do not alter $Ca^{2+}$-dependent inactivation. (A) Cartoon depicting the locations of the C-terminal truncation mutants relative to the putative intracellular motifs. (B) Representative PKD2-L1 currents and the time courses of the potentiation and inactivation for the truncation mutants relative to the Wt channel. Currents were recorded in physiological $[Ca^{2+}]$ (same conditions as described in *Figure 2*). (C) Representative currents activated by +10 mV voltage steps from -100 to +100 mV from the Wt and truncated PKD2-L1 channel. (D) Corresponding voltage dependence of the kinetics of channel opening ($\tau_{peak}$) and channel closure ($\tau_{tail}$). (E) The average PKD2-L1 channel current density measured at +100 mV.

The following figure supplement is available for figure 4:

**Figure supplement 1.** Cell surface expression of the non-functional PKD2-L1 truncation mutants.

configuration and measured the outward single channel events in response to uncaged internal calcium (*Figure 3*). Immediately upon uncaging $Ca^{2+}$-bound NP-EGTA by a 1 s, 405 nm laser pulse, PKD2-L1 open probability ($P_o$) was reduced by 27-fold. The channel was identified as PKD2-L1 since its conductance was consistent with PKD2-L1 homomers (outward 198 pS, inward 121 pS (*DeCaen et al., 2013*), the conductance was not observed in untransfected HEK-293T cells, and the single channel conductance and macroscopic whole cell currents were blocked by dibucaine (*Figure 1—figure supplement 1*). These results demonstrate that PKD2-L1 is long-term-inactivated at high internal $[Ca^{2+}]$ and that this process can occur independent of detectable channel potentiation.

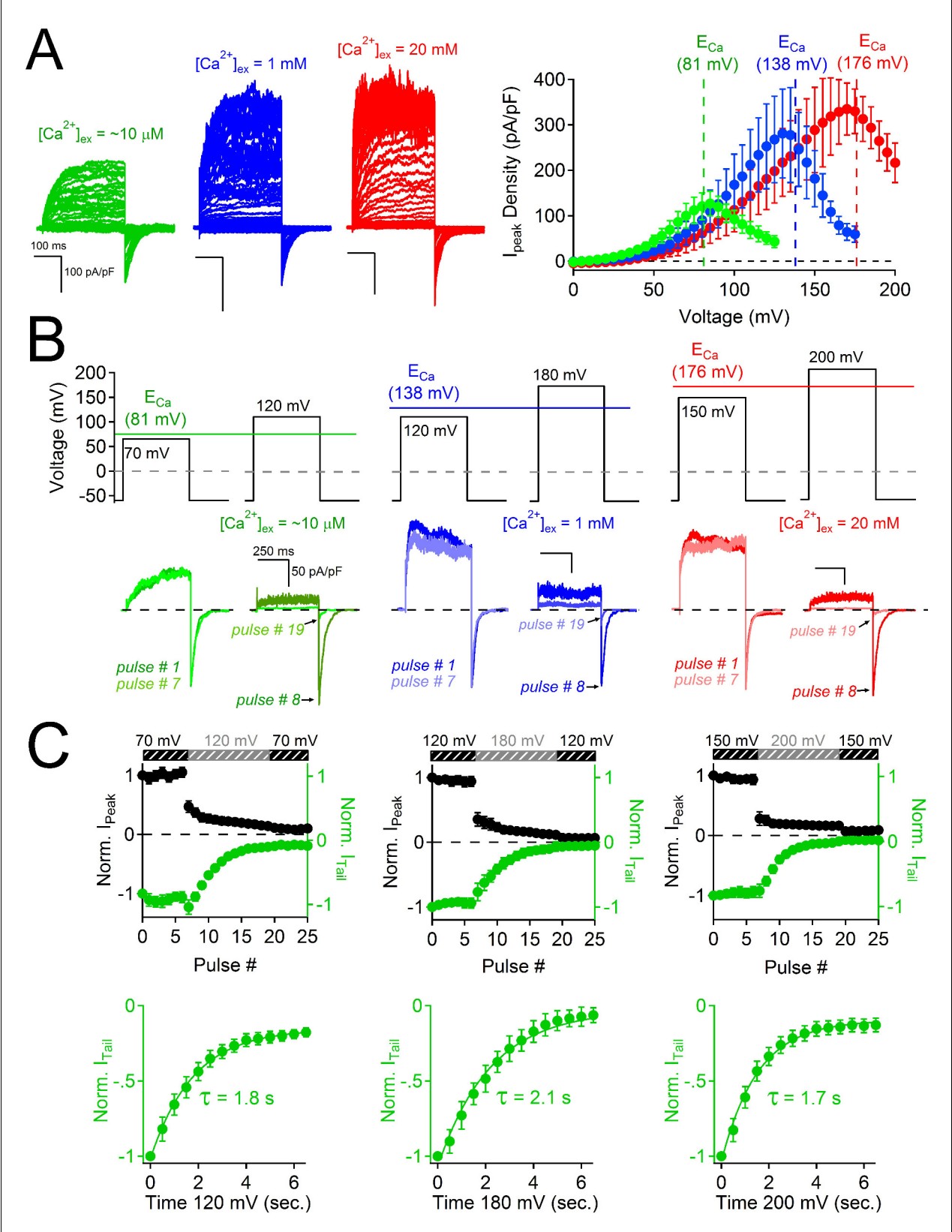

**Figure 5.** Block of PKD2L-1 by outward Ca$^{2+}$ triggers channel inactivation. (A) *Left*, Exemplar currents measured with the indicated [Ca$^{2+}$]$_{ex}$ (inset). *Right*, resulting outward current densities at the indicated E$_{Ca}$ (N = 6–7, Error ± SEM). (B) PKD2-L1 currents recorded from separate cells activated in the *Figure 5 continued on next page*

*Figure 5 continued*

presence of 10 μM, 1mM and 20 mM $[Ca^{2+}]_{ex}$ by a series of depolarizations (0.2 Hz) hyperpolarized and depolarized relative to $E_{Ca}$. (**C**) *Top*, resulting normalized peak and tail currents (Error ± SEM, N =4–5 cells). *Bottom*, PKD2-L1 inactivation rate in each $[Ca^{2+}]_{ex}$ condition (10 μM, 1 mM and 20 mM) was determined by single exponential fitting of the decay of the tail current amplitude ($\tau_{inact.}$ = 1.8 ± 0.2 s; 2.1 ± 0.2 s; 1.7 ± 0.2 s, respectively).

The following figure supplement is available for figure 5:

**Figure supplement 1.** Outward PKD2-L1 current is blocked at membrane potentials positive to $E_{Ca}$.

## PKD2-L1 channel C-terminal EF-hand domains do not mediate inactivation

What is the mechanism by which internal $Ca^{2+}$ inactivates the PKD2-L1 channel? The C-terminus of PKD2-L1 contains an oligomerization domain (OD, 588–608) (*Chen et al., 2015*), putative EF-hands (633–665) (*Li et al., 2002*) and a coiled-coil domain (CC, 690–737) (*Yu et al., 2012*). We generated truncations above and below each of the C-terminal motifs to determine their effect on channel function (*Figure 4A*). We found that the most drastic truncation, located before the first coiled-coil domain (Stop-588), abolished PKD2-L1 channel function (*Figure 4C*). Surface expression of the 588-Stop C-terminal cleavage mutant was unaltered in comparison to Wt PKD2-L1, as assayed by membrane biotinylation (*Figure 4—figure supplement 1*). The intact OD might be critical for channel function or be a subunit oligomerization site, as previously proposed (*Zheng et al., 2015*). However, truncating the PKD2-L1 channel just after the OD domain (Stop-608), or before and after the EF-hands and coiled-coil domain, did not alter normal current density or voltage dependence (*Figure 4B–E*). With the exception of the 508-stop truncation, all truncations were potentiated, then inactivated in normal intracellular $[Ca^{2+}]$ (*Figure 4B*). Thus the EF-hands and coiled-coil domain do not appear to be required for PKD2-L1's calcium-dependent potentiation or inactivation (nor for potentiation by calmidazolium, see below).

## PKD2-L1 channels are inactivated after $Ca^{2+}$ block

The results above exclude the possibility that PKD2-L1's cytoplasmic EF-hands and the coiled-coil domains are involved in inactivating its current or in trafficking the channel to the membrane. When measuring the voltage dependence of the PKD2-L1 channel under physiological $Ca^{2+}$ conditions, we observed that PKD2-L1 channels exhibit a biphasic current-voltage relationship, in which the outward current decreases at potentials beyond the reversal potential for calcium ($E_{Ca}$) (*Figure 5A*). By altering extracellular calcium ($[Ca^{2+}]_{ex}$), we noted that net nonselective outward current always declined positive to $E_{Ca}$ (*Figure 5A*). Repetition of the protocol demonstrated that PKD2-L1 channels had completely inactivated; that is, PKD2-L1 currents could no longer be elicited during the recording period (*Figure 5—figure supplement 1*). This data suggests that outward PKD2-L1 currents were stably blocked and/or inactivated by outward moving $Ca^{2+}$ ions. We tested TRPM7, TRPV1, and TRPV3 for similar behavior and found that none of their outward currents exhibited this biphasic dependence (*Figure 5—figure supplement 1*, TRPV1 data not shown). Comparing the PKD2-L1 outward current to the inward tail current elicited by repolarization to −60 mV (*Figure 5—figure supplement 1*), we observed that the outward current was blocked, whereas the inward tail current remained relatively constant. These observations suggest that $Ca^{2+}$ induced block and subsequent inactivation of all outward current, while inward flux was unimpeded. To examine this phenomenon, we used a voltage protocol designed to capture the rates of $Ca^{2+}$- dependent block and inactivation (*Figure 5B,C*). A series of voltage clamp steps 10–30 mV negative to $E_{Ca}$ were applied so that inward tail currents and outward currents were stable over time. Thus, inward $Ca^{2+}$ could permeate PKD2-L1 without blocking or inactivating the channel. Then, the depolarization step was increased to 28–42 mV more positive than $E_{Ca}$, where the outward currents were immediately blocked by outwardly moving calcium ions. In contrast, tail currents triggered by repolarization to -60 mV were not immediately blocked, but progressively decreased after each depolarization positive to $E_{Ca}$, indicating that the inward PKD2-L1 currents were not long-term- inactivated until after the outward current was blocked by $Ca^{2+}$. By plotting the sum of the durations of the depolarization steps positive to $E_{Ca}$ against the tail current magnitude (*Figure 5B,C*), we estimated the rate of inactivation, $\tau_{inact} \approx$

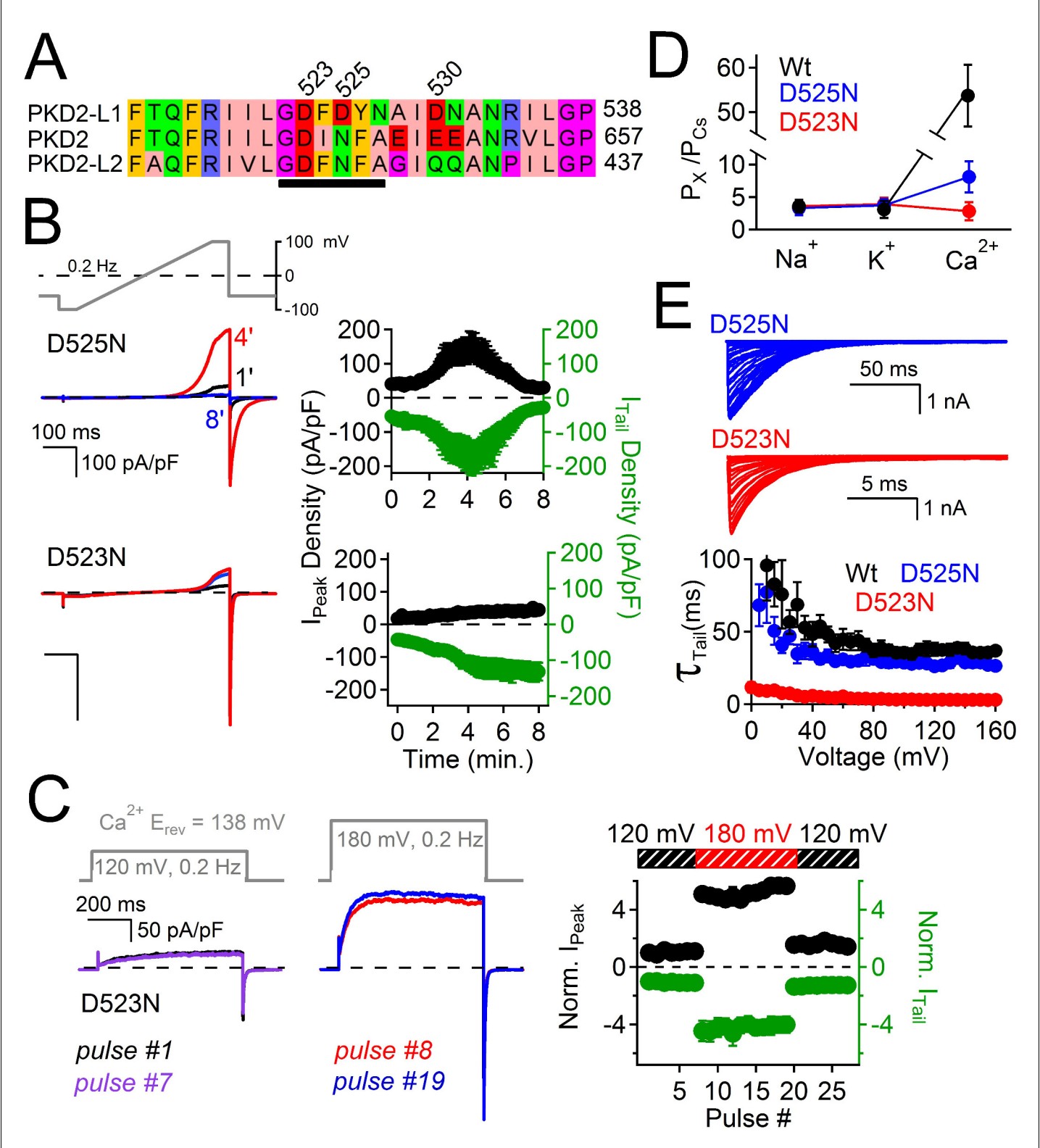

**Figure 6.** Filter mutant D523N is not blocked by outward $Ca^{2+}$ and is not inactivated. (**A**) Amino acid alignment of the pore regions of PKD2 family members. The black bar indicates the selectivity filter. (**B**) Representative currents and time course of the average tail and peak current densities for the selectivity filter mutants (Error ± SEM, N =7 cells). (**C**) *Left,* representative D523N currents activated by a series of depolarizations (0.2 Hz) negative (+120 mV) and positive (+180 mV) to $E_{Ca}$ (138 mV). *Right,* resulting normalized peak and tail current amplitudes (Error ± SEM, N =5 cells). (**D**) The relative
*Figure 6 continued on next page*

*Figure 6 continued*

permeability ($P_x$/$P_{Cs}$) of Na$^+$, K$^+$ and Ca$^{2+}$ for Wt and mutant channels. These values were calculated using the measured reversal potentials from the steady state voltage-current relationships in *Figure 6—figure supplement 1*, and tabulated in *Figure 6—source data 1*). (E) *Top*, representative tail currents activated by repolarizing -60 mV from the indicated potentials. *Bottom*, Corresponding voltage dependence of the channel closing (deactivation) kinetics ($\tau_{tail}$;Error ± SEM, N=6–8).
The following source data and figure supplement are available for figure 6:

**Source data 1.** A table listing the relative permeabilities of cations through the PKD2-L1 channels as estimated by the measured reversal potentials ($E_{rev}$).
**Figure supplement 1.** The current-voltage relationships of PKD2-L1 channels in the presence of different extracellular cations.

2 s (1.8 ± 0.2 s; 2.1 ± 0.2 s; 1.7 ± 0.2 s for the [Ca$^{2+}$]$_{ex}$ conditions 10 μM, 1 mM and 20 mM respectively).

## Selectivity filter residues mediating selectivity and block

The above results suggest that outward-going calcium ions through PKD2-L1 block the pore and induce a long period of channel inactivation. Thus, we sought to reduce the affinity for the Ca$^{2+}$ coordination sites within the selectivity filter to relieve the outward current block by Ca$^{2+}$ and thus reduce inactivation. We previously reported that the PKD2-L1 conductance could be abolished by double serine or alanine mutations, to Asp523 and Asp525 (D523S:D525S or D523A:D525A), whereas mutation of Asp530 (D530N) did not alter the conductance (*DeCaen et al., 2013*). We also tested individual alanine substitutions to either position (data not shown), which also abolished the PKD2-L1 conductance. These data suggest that residues D523 and D525 are critical positions within the PKD2-L1 selectivity filter. Since D523 is the most conserved residue among the PKD2 family members, we propose that it is essential for block, inactivation, and its modest Ca$^{2+}$ selectivity (*Figure 6A*). Since asparagine (Asn, N) has a similar volume (≈ 11 nM (*Nauli and Zhou, 2004*)) but its carboxamide side chain is not as electronegative as the carboxylate found in the native aspartate (Asp, D), we generated individual substitutions of Asn for D523 and D525. We found that at similar current densities, D523N mutant-associated currents did not inactivate over an 8 min time course, whereas D525N currents first potentiated and then inactivated similar to the Wt channel (*Figure 6B*). To determine if the D523N mutations altered outward Ca$^{2+}$-dependent inactivation, we used the same depolarization protocol as used in *Figure 5B* and observed that the outward peak and inward tail currents were stable over time and potential, indicating that Ca$^{2+}$ does not block or inactivate the D523N current (*Figure 6C,D*). Furthermore, the voltage dependence of the D523N outward current is not biphasic at potentials more positive to E$_{Ca}$ (*Figure 6—figure supplement 1*).

These data demonstrate that Ca$^{2+}$ does not block the D523N current in the outward direction and thus the ensuing inactivation does not occur. The selectivity for Ca$^{2+}$ of the filter mutant channels D523N and D525N are reduced 19- and 8- fold (respectively) compared to the wild type channel ($P_{Ca}$/$P_{Cs}$ = 7), as estimated by the change in E$_{rev}$ when measured in Na$^+$, K$^+$, Ca$^{2+}$ and Cs$^+$ extracellular conditions (*Figure 6D*, *Figure 6—figure supplement 1*, *Figure 6—source data 1*). The speed of tail current decay of the D523N channel was enhanced ≈ 20-fold (measured at −100 mV, Wt $\tau_{Tail}$ = 43 ms ± 3; D523N $\tau_{Tail}$ = 2.3 ms ± 0.2), whereas those from the D525N mutation were much less affected (D525N $\tau_{Tail}$ = 37 ms ± 2; *Figure 6E*).

Despite dramatically altering PKD2-L1 kinetics and selectivity, the D523 mutation retains sensitivity to modulation by the membrane-permeant calmodulin antagonist, calmidazolium (Wt EC$_{50}$ = 2.6 μM ± 6; D523N EC$_{50}$ = 9 μM ± 7; also see DeCaen et al. [*DeCaen et al., 2013*]) and block by the amide local anesthetic, dibucaine (Wt IC$_{50}$ = 31 μM ± 5; D523N IC$_{50}$ = 23 μM ± 3; *Figure 7—figure supplement 1*, *Figure 7—source data 1*). These results suggest that modulation by calmidazolium and block by dibucaine are preserved in the D523N mutant, suggesting that these compounds alter PKD2-L1 function through sites allosterically coupled to the filter. Together, these results demonstrate that while both D523 and D525 participate in Ca$^{2+}$-selectivity of the pore, position D523 is the most critical to Ca$^{2+}$- inactivation.

Since Asp523 apparently forms the high affinity coordination site for Ca$^{2+}$ that is responsible for Ca$^{2+}$-selectivity and initiates outward Ca$^{2+}$-dependent inactivation, we hypothesized that it might

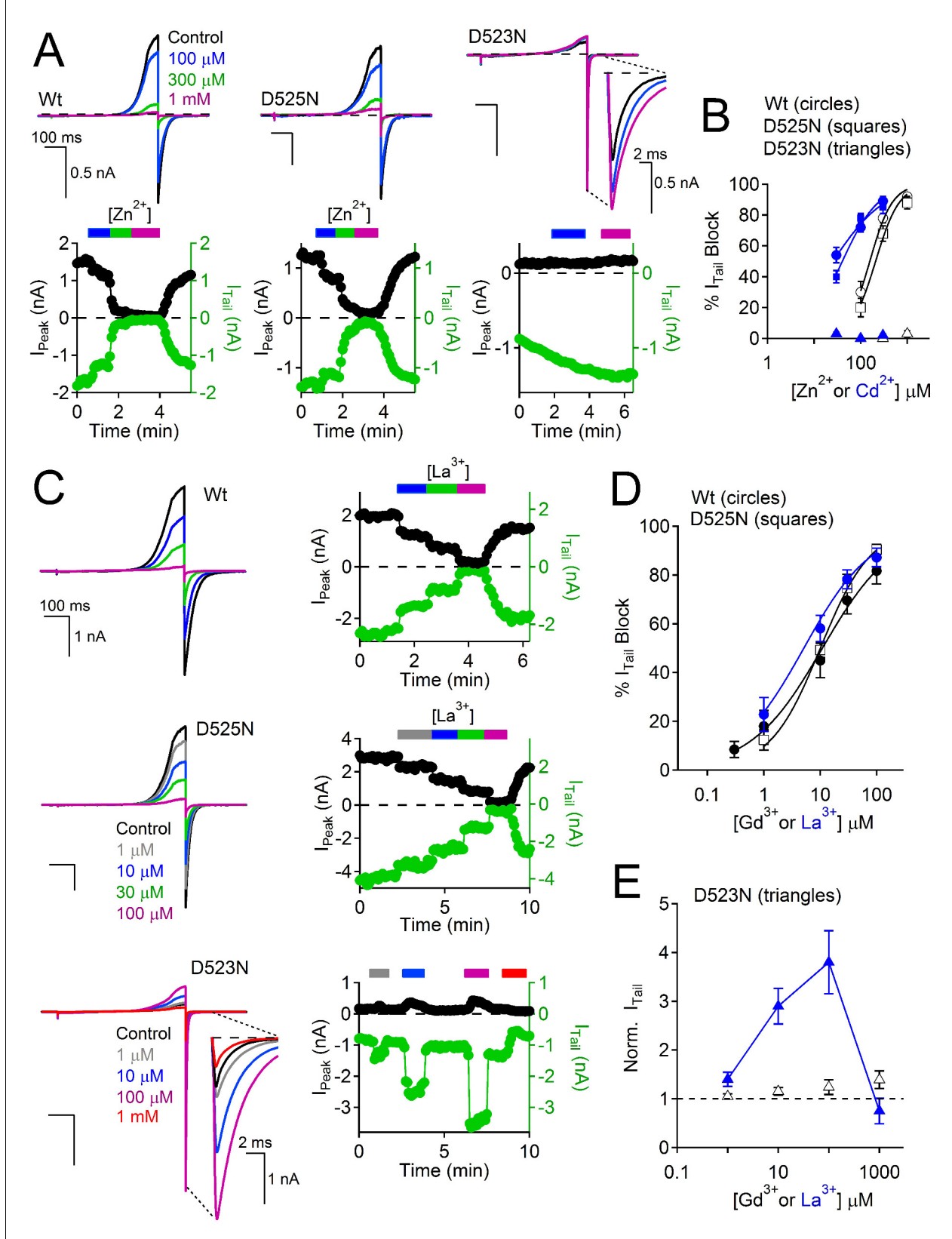

**Figure 7.** Loss of transition metal block in selectivity filter mutant, D523N. (**A**, **B**) Effects of $Zn^{2+}$ and $Cd^{2+}$ on PKD2-L1 currents. *Top*, Exemplar PKD2-L1 currents activated by voltage ramps at the indicated $[Zn^{2+}]$. *Bottom*, Corresponding time courses of the $Zn^{2+}$ block of the Wt and D525N PKD2-L1

*Figure 7 continued on next page*

*Figure 7 continued*

channels. The internal buffer was 15 mM BAPTA and 5 mM EGTA, which prevented $Ca^{2+}$ accumulation and inactivation of the PKD2-L1 currents. (B) Divalent metal concentration-dependent block of the Wt, D523N and D525N channels (Error ± SEM, N = 4–6 cells). (C–E) Effects of trivalent metal block. *Left*, Exemplar PKD2-L1 currents activated by voltage ramps in the presence of $La^{3+}$. *Right*, Corresponding time courses of the $La^{3+}$ block of the indicated PKD2-L1 channels. (D) Concentration-dependent block of PKD2-L1 and D525N channels by trivalent ions, and (E) potentiation of the D523N filter mutant channel (Error ± SEM, N = 4–5 cells).

The following source data and figure supplement are available for figure 7:

**Source data 1.** A table listing the potencies ($IC_{50}$) of PKD2-L1 current antagonism by dibucaine and transition metals.

**Figure supplement 1.** Calmidazolium activation and dibucaine block is preserved in the D523N filter mutant channel.

also be the binding site for divalent metals and trivalent metal antagonists. The transition metals, cadmium and zinc, blocked the PKD2-L1 Wt current ($IC_{50}$ = 25 μM ± 5; $IC_{50}$ = 156 μM ± 15) and the D525N filter mutant ($IC_{50}$ = 40 μM ± 8; $IC_{50}$ = 209 μM ± 28), but were ineffective antagonists of the D523N channel at concentrations below 1 mM (*Figure 7A,B*; *Figure 7—source data 1*). Trivalent cations, such as lanthanum and gadolinium, are commonly used non-specific blockers of $Ca^{2+}$-permeable channels. We found that the PKD2-L1 channel is 20-times more sensitive to block by $Gd^{3+}$ ($IC_{50}$ = 9 μM ± 3) and $La^{3+}$ ($IC_{50}$ = 3 μM ± 4) than reported for members of the TRPV and TRPA families ($IC_{50} \geq$ 200 μM, *Figure 7C,D*) (*Leffler et al., 2007*; *Banke, 2011*; *Xu et al., 2002*). This potency was preserved in the D525N filter mutant channel ($IC_{50}$ = 11 μM ± 3 and 1 μM ± 4, respectively). However, the D523N channel displayed anomalous mole fraction behavior in the presence of $La^{3+}$, conducting through the channel at low to mid-μM concentrations (1–100 μM), but blocking the basal current in the millimolar range (*Figure 7E*). These findings suggest that D523 forms a coordination site for divalent and trivalent metal cations.

## Discussion

Regulation of ion channels by cytoplasmic calcium is a common theme in biology (*Clapham, 2007*; *Yu and Catterall, 2004*) and can occur through direct binding to the channel protein or through adaptor or modulatory proteins. Calcium binding proteins like calmodulin often bind to $Ca^{2+}$-permeant ion channels and alter their function (*Ben-Johny et al., 2015*). Calcium coordinating motifs within channel proteins, such as cytoplasmic EF-hands found in voltage gated $Ca_V$ channels, alter channel inactivation based on localized calcium accumulation. Here, we have shown a direct mechanism of calcium regulation of the PKD2-L1 channel, in which outward $Ca^{2+}$ binds in the selectivity filter and initiates inactivation of the channel. Calcium and other ions pass inwardly through the PKD2-L1 channel as long as the electrochemical gradient permits; when channel densities are high, this is sufficient to significantly shift internal ion concentrations as seen by the slowly shifting $E_{rev}$. However, depolarizations above $E_{Ca}$ lead to channel block and inactivation (as proposed in *Figure 8A*). To our knowledge, this is the only reported ion channel with $Ca^{2+}$-dependent rectification due to pore block-inactivation, and so far is unique to PKD2-L1.

Many members of the TRPA, C, M and V families are potentiated and inactivated by $Ca^{2+}$ dependent mechanisms (*Gordon-Shaag et al., 2008*; *Zhu, 2005*). Thus far, the proposed mechanism(s) by which $Ca^{2+}$ inhibits TRP channels are indirect, such as with $PI(4,5)P_2$-depletion, $Ca^{2+}$-regulated kinases, phosphatases, and phospholipases, via both calmodulin-dependent and independent processes (*Gordon-Shaag et al., 2008*; *Zhu, 2005*). We showed that $Ca^{2+}$ occupancy in the filter blocks the outward PKD2-L1 current and triggers its long-term inactivation. Although mutating the selectivity filter of TRPA1 (D918) can abolish extracellular $Ca^{2+}$-dependent potentiation and inactivation, the effect of extracellular $Ca^{2+}$ on these processes follows elevation of intracellular calcium and is proposed to operate through an indirect, unknown mechanism (*Wang et al., 2008*). For PKD2-L1, outward $Ca^{2+}$ current inactivates the channel after binding within the selectivity pore. Although we cannot rule out allosteric modulation by calmodulin or kinases during inactivation, involvement of the C-terminal EF-hand and coiled-coil are unlikely since no effect was observed on inactivation when these motifs were removed. Likewise, for the TRPA1 channel, removing the EF-hands did little to alter the onset of potentiation and inactivation (*Wang et al., 2008*). Our observations from our

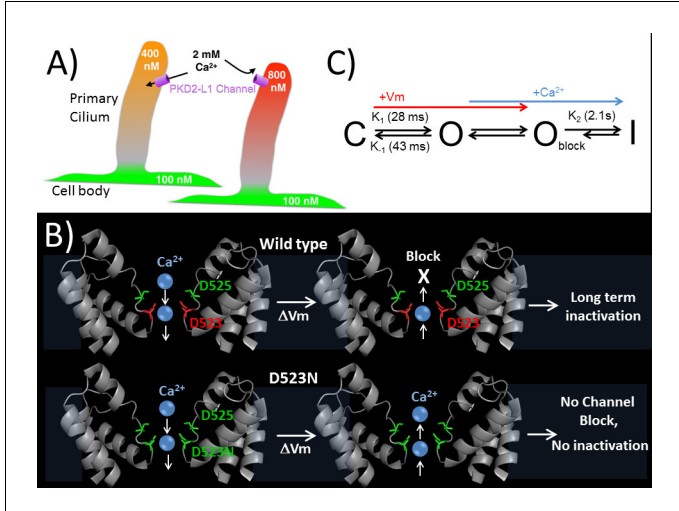

**Figure 8.** Proposed kinetic scheme of PKD2-L1 channel states and a hypothetical model of $Ca^{2+}$ coordination sites in the selectivity filter. (**A**) Calcium clamp of restricted spaces, such as primary cilia, by PKD2-L1. Resting $[Ca^{2+}] \cong$ 500–700 nM in primary cilia. PKD2-L1 channels in the cilia membrane are potentiated before inactivation by $[Ca^{2+}]$ < ~500 nM, but simply inactivated at $[Ca^{2+}]$ > ~700 nM. (**B**) Model of Wt and D523N filters with proposed $Ca^{2+}$ movement in response to membrane potential. The PKD2-L1 low affinity $Ca^{2+}$ (D525) and high affinity (D523) binding sites are colored yellow and red, respectively, corresponding to residues N921 and D918 in the TRPA1 structure (PDB: 3J9P) as a substitute for the undetermined PKD2-L1 structure. (**C**) Proposed four channel state scheme C = closed; O = open; $O_{block}$ = open channels blocked by outwardly moving $Ca^{2+}$ and I = inactivated (long-term). Corresponding rate constants (K) are indicated between the three channel states ($K_1$ = opening, $K_{-1}$ = closure, $K_2$ = inactivation). The colored arrows indicate the direction of channel state stimulus by either membrane depolarization ($+V_m$) or outwardly moving calcium ions ($Ca^{2+}$). The recovery from calcium-dependent inactivation is shown as a shortened arrow since inactivation is irreversible on the time scale of the experiments (<20 min).

PKD2-L1 EF-hands truncation are in agreement with results from the PKD2-L1 splice variants expressed in the liver, where a naturally occurring C-terminal EF-hand truncation is found to have no effect on $Ca^{2+}$-dependent inactivation (*Li et al., 2002*). Thus if C-terminal modulation is involved, the location for binding of a modulator would have to be located above the oligomerization domain, which appears to be critical for PKD2-L1 function, as the trafficking of the Stop-588 mutant was the same as the full length channel.

We have shown that even potent (BAPTA, 5 mM) cytoplasmic $Ca^{2+}$ buffers can be saturated when PKD2-L1 channels are overexpressed. Once occupied by outwardly moving calcium ions, the PKD2-L1 channels irreversibly inactivates in seconds ($\tau$ = 2.1 s). Thus our results explain the inactivation phenomenon observed in Chen et al. (*Chen et al., 1999*) and describe a direct level of $Ca^{2+}$ feedback inhibition of the PKD2-L1 channel. Given that cytoplasmic $[Ca^{2+}]$ ranges from 50 nM to as high as 10 μM (near mouths of channels), what is the physiological relevance of outward calcium current regulation in PKD2-L1 channels? Because of the cell's large volume (typically 1–3 pL), it is unlikely that the concentration of calcium in the cytosol would ever reach sufficient concentrations to move outwardly through PKD2-L1 channel and trigger inactivation. However, because PKD2-L1 channels have large inward conductances (PKD2-L1: 120–150 pS and PKD1-L1 +PKD2-L1 = 86 pS) (*DeCaen et al., 2013*; *Chen et al., 1999*; *Shimizu et al., 2009*) with modest selectivity for $Ca^{2+}$ ($P_{Ca}/P_{Cs}$ PKD2-L1 and PKD1-L1 +PKD2-L1 = 6–15) (*DeCaen et al., 2013*; *Shimizu et al., 2009*), the expression of just a few channels (1–10) in small cellular compartments could result in large changes in local $Ca^{2+}$ concentrations. For example, the homomeric PKD2-L1 is enriched in the membrane of the dendritic nob of CSF contacting neurons, which have small volumes (113 fL) (*Orts-Del'Immagine et al., 2014*; *Orts-Del'Immagine et al., 2016*). Here, PKD2-L1's alkaline-pH-stimulated channel activity ($P_o$ = 0.02) would increase the dendritic nob cytoplasmic calcium by $\approx$ 22 μM per channel opening. In smaller compartments, like the primary cilium (0.5 fL), where roughly 30 heteromeric PKD1-L1 +PKD2-L1 channels are expressed, ATP stimulation ($P_o$ = 0.01) would increase the

cilioplasmic calcium by $\approx$ 31 µM per channel opening. Thus, prolonged PKD2-L1 (or PKD1-L1 +PKD2-L1) channel openings could result in increases in cellular compartment [Ca$^{2+}$] up to the external [Ca$^{2+}$]. Most important, we hypothesize that PKD2-L1 maintains Ca$^{2+}$ concentration in the 500 - 700 nM range observed in primary cilia (*Nelson et al., 2010*; *Wu et al., 2000*). As shown in *Figure 2*, entering Ca$^{2+}$ transiently potentiates PKD2-L1 up to ~750 nM [Ca$^{2+}$]$_{In}$ - above this level, inhibition dominates. This behavior would serve as [Ca$^{2+}$] clamp, a feedback mechanism that counteracts Ca$^{2+}$ loss via diffusion into the cytoplasm (*Figure 8B*). This mechanism would ensure that the primary cilium is a specialized calcium compartment, even though no calcium diffusion barrier exists between cilium and cytoplasm (*Nelson et al., 2010*). However, the Ca$^{2+}$ exchangers and intrinsic buffers present in each compartment are not known, limiting a more detailed interpretation of this mechanism.

We have established that D523 within the filter is responsible for Ca$^{2+}$-selectivity as ions pass inward, and for block by Ca$^{2+}$ as it moves out of the cell (*Figure 6*, *Figure 6-figure supplement 1*). Independent substitutions of the aspartate filter residues 523 and 525 with asparagine resulted in a significant reduction of Ca$^{2+}$-selectivity, while independent alanine substitutions for either residue resulted in a non-conducting channel. Based on these observations we propose that this unique calcium conductance is achieved with two ion coordinating sites involving filter residues D523 and D525. We speculate that Ca$^{2+}$ unidirectional conductance is achieved by a 'knock-on' ionic interaction, where the outer residue (D525) coordinates Ca$^{2+}$ with weaker affinity than the inner residue (D523). In this model, the interaction of Ca$^{2+}$ with D523 is broken by repulsion by a second Ca$^{2+}$ occupying the outer D525. When the membrane potential is more depolarized than E$_{Ca}$, Ca$^{2+}$ occupies the high-affinity binding site (D523). Since there is no inner Ca$^{2+}$ coordinating site to electrostatically 'knock-off' Ca$^{2+}$ from D523, it is not displaced from this site and outward current is blocked. Since Ca$^{2+}$ does not block the D523N channel, we propose that Ca$^{2+}$ is able to move in both directions in the mutant channel's selectivity filter (*Figure 8B*). This effect can be explained by the loss of the high-affinity binding site within the innermost portion of the selectivity filter. Since D523 is conserved within the PKD2 family, it is possible that other members of PKD2 family may share this same unidirectional calcium conductance, although the selectivities of homomeric PKD2 and PKD2-L2 channels have yet to be defined.

We established that D523 is the metal binding site for divalent and trivalent ions within the selectivity filter of PKD2-L1. The potency (IC$_{50}$) of PKD2-L1 block by Gd$^{3+}$ was shifted from 9 µM to greater than 1 mM in the D523N channel, which reflects a loss of affinity >6.7 kcal/mol. Conversely, the D523N mutant conducts the smaller trivalent La$^{3+}$ at µM concentrations, whereas the Wt and D525N channels are blocked within this concentration range. These seeming incompatible observations are reminiscent of La$^{3+}$ anomalous mole fraction effects reported in Wt members of the TRPC family (C3, C4, C5 and C6) (*Jung et al., 2003*; *Strübing et al., 2001*; *Hofmann et al., 1999*). Thus, whether trivalent and divalent metals permeate or block TRP channels is likely determined by the strength of their electrostatic interactions with acidic residues that line the selectivity filter.

A simple preliminary four state model of PKD2-L1 is shown in *Figure 8C*. We have defined the resting state found at negative potentials prior to depolarization as the closed state (C), although PKD2-L1 has some of constitutive activity even at −100 mV (P$_O$ = 0.007). After depolarization, our analysis of macroscopic and single channel currents demonstrates that channel openings are transiently increased upon hyperpolarization (*Figure 1—figure supplement 1*). We proposed that these large, slowly decaying tail currents represent the population of channels transitioning from the open (O) to closed (C) state. The closed to open transition (K$_1$) can be measured by the kinetics of the peak currents during membrane depolarization (<10 mV). While it is possible that PKD2-L1 gating may have voltage dependent inactivation properties akin to some delayed rectifying potassium channels (I$_{Kr}$) (*Warmke and Ganetzky, 1994*; *Trudeau et al., 1995*), our experiments did not capture this kinetic property. Further biophysical characterization will be necessary to test this possibility. The PKD2-L1 channel opening (K$_1$) and closure (K$_{-1}$) are reversible and modestly voltage dependent, increasing 1.5x and 2.2x with 100 mV membrane potential shifts, respectively. Outwardly moving Ca$^{2+}$ rapidly block the open channel state (O$_{block}$) and trigger a slower long-term inactivation process (K$_2$). The rate of inactivation (K$_2$) is dependent on outward Ca$^{2+}$ current and is irreversible on the time scale of the experiments. Interestingly, the enhanced onset of channel opening and closing found in the D523N mutant may be related to the lack of Ca$^{2+}$ selectivity and Ca$^{2+}$-dependent inactivation. These observations demonstrate that the rates of PKD2-L1 gating can be altered by mutating positions within the selectivity filter, suggesting that permeant ion coordinating

sites are partly involved. This feature has precedence in BK channels, where permeant ions alter the opening and closing transitions but do not alter the calcium- or voltage-dependent activation pathways (*Thompson and Begenisich, 2012*; *Piskorowski and Aldrich, 2006*). It is possible that PKD2-L1 pore blockade by outwardly moving $Ca^{2+}$ ion(s) are displaced by inwardly $Na^+$ current and a slow $Ca^{2+}$ unbinding process prolongs the time course of channel closure. That is, hyperpolarization helps clear $Ca^{2+}$ ions from the PKD2-L1 selectivity filter. This effect may be analogous to displacement of quaternary ammonium ion(s) by inward $K^+$ current conducted by $K_V$ channels found in the squid giant axon (*Armstrong, 1971*).

## Materials and methods

### Whole-cell voltage-clamp experiments

HEK 293T cells were transiently transfected with the mammalian cell expression plasmid pTracer CMV2 containing the human PKD2-L1 gene (isoform 1). Cells were seeded onto glass coverslips and placed in a perfusion chamber, enabling changes in extracellular conditions. Data generated from cells patched in the whole cell configuration with <1 GΩ of resistance and >90 pA of leak current ($V_{mem}$ =-60 mV) were not analyzed due to insufficient voltage control. Cell-attached single channel data with seal resistance <8 GΩ were not analyzed. Unless otherwise indicated, the pipette electrode solution contained (in mM): CsMES (80), NaCl (20), HEPES (10), $MgCl_2$ (2), $Cs_4$-BAPTA (5); $CaCl_2$ was added to achieve 90 nM free $Ca^{2+}$ and pH was adjusted to 7.4 with CsOH (MaxChelator (*Bers et al., 2010*)). The standard bath solution contained NaCl (150), HEPES (10), $CaCl_2$ (2) and pH was adjusted with NaOH. When testing the relative permeability of monovalent cations, the bath solution contained (in mM): X-Cl (150), HEPES (10) and the pH was adjusted with X-OH; X is the indicated monovalent cation. When testing the relative permeability of $Ca^{2+}$, the bath solution contained: NMDG (100), $CaCl_2$ (20), HEPES (10), and pH was adjusted with $CaOH_2$. All saline solutions were adjusted to 300 mOsm (±5) with mannitol, if needed. Data was analyzed by Igor Pro 7.00 (Wavemetrics, Lake Oswego, OR). The reversal potential, $E_{rev}$ was used to determine the relative permeability of monovalent cation X to $Cs^+$ ($P_X/P_{Cs}$) according to the following equation:

$$\frac{P_x}{P_{Cs}} = \frac{\alpha_{Cse}}{\alpha_{xe}} \left[ \exp\left(\frac{\Delta E_{rev}}{RT/F}\right) \right]$$

where $E_{rev}$, $\alpha$, R, T and F are the reversal potential, effective activity coefficients for cation **x** (i, internal and e, external), the universal gas constant, absolute temperature, and the Faraday constant, respectively. Based on out measurements of the $E_{rev}$ under our semi-bi-ionic conditions, the PKD2-L1 permeability of $Cs^+$ and $NMDG^+$ ions are equivalent, but are much less permeant to other cations tested ($Ca^+$, $K^+$ and $Na^+$). The effective activity coefficients ($\alpha_x$) were calculated using the following equation:

$$\alpha_x = \gamma_x [X]$$

where $\gamma_x$ is the activity coefficient and [X] is the concentration of the ion. For calculations of membrane permeability, activity coefficients ($\gamma$) were calculated using the Debye-Hückel equation: 0.74, 0.72, 0.69, and 0.29 correspond to $Na^+$, $K^+$, $Cs^+$ and $Ca^{2+}$, respectively. To determine the relative permeability of divalent cations to $Cs^+$, the following equation was used:

$$\frac{P_x}{P_{Cs}} = \frac{\left\{ \alpha_{Csi} \left[ \exp\left(\frac{E_{rev}F}{RT}\right) \right] \left[ \exp\left(\frac{E_{rev}F}{RT}\right) + 1 \right] \right\}}{4\alpha_{xe}}$$

$E_{rev}$ for each cation condition was corrected to the measured liquid junction potentials (-4.4 to 3.4 mV). Single channel events (*Figure 1—figure supplement 1* and *Figure 3*) were measured with standard extracellular saline in the pipette and high potassium saline in the bath (in mM): KCl (125), NaCl (20), HEPES (10) and $CaCl_2$ (2) to neutralize the resting membrane potential. To determine the time course of channel opening ($\tau_{peak}$) and closure ($\tau_{tail}$) the peak current during the depolarization ($\triangle$ mV) and the tail currents during repolarization ($-60$ mV) were fit using the following exponential equation:

$$f(x) = B + A\,exp\left[\left(\frac{1}{\tau}\right)x\right]$$

where $\tau$ is the time constant. The internal accumulation of calcium during experiments in *Figure 2—figure supplement 1* was estimated using the following equation:

$$= Q \times r \times \frac{charge}{1.6x10^{-19}Coulombs} \times \frac{Ca^{2+}}{2\,charges} \times \left(6.02x10^{23}mol\right)^{-1} \times \left(2.14x10^{-12}\,L\right)^{-1}$$

where Q is the cumulative integrated $I_{tail}$ current and r (0.199) is the product of the extracellular calcium (2 mM) to sodium (150 mM) ratio multiplied by the relative permeability of calcium for sodium ions ($P_{Ca}/P_{Na}$ = 14.9). With a starting condition of 100 nM free $Ca^{2+}$, the cumulative [free $Ca^{2+}$] was estimated using MaxChelator (*Bers et al., 2010*), where 5 mM BAPTA was used as an internal $Ca^{2+}$ buffer. We assume that the 5 mM BAPTA perfusing the cell was the dominant buffer over intrinsic mobile and immobile buffers which 'typically' bind $Ca^{2+}$ in the range of 40 bound/1 free (*Zhou and Neher, 1993*). HEK293 cells have an average radius of 8 µM and a volume of ~2100 µM (*Nauli and Zhou, 2004*), ($2.1 \times 10$–12 L). Single channel current magnitude and open time was estimated at the resting membrane potentials for PKD2-L1 channel in CSF-contacting neurons ($-55$ mV) (*Orts-Del'Immagine et al., 2016*), PKD2-L1 (*DeCaen et al., 2013*) and PKD1-L1 +PKD2-L1 channels from primary cilia ($-17$ mV) (*Delling et al., 2013*).

## $Ca^{2+}$ uncaging experiments

PKD2-L1 transfected HEK cells were incubated with extracellular o-nitrophenyl EGTA (NP-EGTA; 4 mM) and Fluo-3-AM (4 mM) for 30 min at room temperature. Cells were visualized with an Olympus FV1000 confocal microscope equipped with an SIM scanner. After establishing a high resistance seal using electrodes with 3–5 mΩ pipette resistance in the on-cell configuration, the cell membrane potential was held at +80 mV and cytoplasmic calcium was uncaged by a 500 ms, 405 nm laser pulse. All images were analyzed using ImageJ (NIH) and IgorPro 7 (Wavemetrics).

## Surface biotinylation assay

Biotinylation of cell-surface proteins was performed using EZ-Link Sulfo-NHS-SS-Biotin (Thermo Fisher Scientific, Waltham, MA) according to the manufacturer's instructions. In brief, 48 hr after transfection, HEK 293T cells were washed with PBS, and EZ-Link Sulfo-NHS-SS-Biotin applied to living cultured cells expressing HA, HA-2L1, HA-2L1 (Stop-588) for 30 min at 4°C. Cells were lysed, and the biotinylated proteins were precipitated using streptavidin agarose beads. The eluted proteins were analyzed by immunoblotting.

# Additional information

### Competing interests

DEC: Reviewing editor, *eLife*. The other authors declare that no competing interests exist.

### Funding

| Funder | Grant reference number | Author |
|---|---|---|
| Howard Hughes Medical Institute | | David E Clapham |
| National Institutes of Health | Pathway to Independence (PI) Award (K99/R00) | Paul G DeCaen |

The funders had no role in study design, data collection and interpretation, or the decision to submit the work for publication.

### Author contributions

PGD, XL, Conception and design, Acquisition of data, Analysis and interpretation of data, Drafting or revising the article, Contributed unpublished essential data or reagents; SA, Conception and

design, Drafting or revising the article, Contributed unpublished essential data or reagents; DEC, Conception and design, Analysis and interpretation of data, Drafting or revising the article

## Author ORCIDs

David E Clapham, http://orcid.org/0000-0002-4459-9428

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
