## [Decision Letter]

Thank you for submitting your work entitled "Outward calcium current initiates inactivation of the PKD2-L1 TRP channel" for consideration by *eLife*. Your article has been favorably evaluated by Richard Aldrich as the Senior editor and three reviewers, one of whom is a member of our Board of Reviewing Editors, and another is Ramón Latorre.

The reviewers have discussed the reviews with one another and the Reviewing Editor has drafted this decision to help you prepare a revised submission.

Summary:

This manuscript identifies the mechanism of Ca-dependent desensitization of a recently described cation channel, PKD2-L1. This channel, which is part of the TRP family, and is expressed in the cilia of kidney epithelial cells; its knockout in mice is embryonic lethal, and its dysfunction is associated with kidney disorders. When expressed in HEK cells, it produces ionic currents that are constitutively active but show a non-instantaneous outward rectification. With step depolarizations, the conductance increases over ~100 ms, and repolarization elicits large tail currents. Repeated ramps demonstrate that the channel is sequentially facilitated and then desensitized by Ca. Through a series of electrophysiological and molecular manipulations, the present experiments demonstrate that the desensitization but not facilitation phenomenon results from pore block by internal Ca^2+^ trying to exit through the channel and interacting with D523 in the selectivity filter to block the channel.

Essential revisions:

The reviewers found the manuscript compelling. Primary questions had to do with (a) clarifying the interpretation of the mechanism, either by a more extensive Discussion accompanied by quantitative justifications and/or reanalysis to clarify the ambiguous points or by direct biophysical modeling (described in detail in points 1 and 2 below), (b) presentation of figures in a way to ensure that key data are all present in the main text (point 3 below), (c) revisiting the Ca permeability analysis (point 4 below (d) clarification of the single channel analysis (point 5 below). Please revise the manuscript accordingly.

Detailed explanation of essential revisions from reviewers' comments.

1) Results/Figure 3: "…the outward current was blocked, whereas the inward tail current remained relatively constant. These observations suggest that Ca^2+^ induced block and subsequent desensitization of all outward current, while inward flux was unimpeded." It is not clear to me how the block can be cumulative over repeated trials (steps here or ramps elsewhere) if inward current is unimpeded. This seems self-contradictory. The authors provide ample evidence that outwardly flowing Ca blocks the channel, but they do not state explicitly what happens to the bound Ca over time (upon repolarization). The data with the D523N mutant in Figure 6—figure supplement 1 shows that the lack of Ca block correlates with an extreme shortening of the long, broad tail current that characterizes the wild type. It therefore seems possible that the blocking Ca (in the wild type) is displaced by inwardly flowing ions, at least in a subset of channels, and a slow unbinding process prolongs the time course of the tail (as in displacement of quaternary ammonium ions by inward K tails in Armstrong 1971). It is hard to tell whether this idea is considered or favored or ruled out by the authors. In the Discussion there is a consideration of knock-on and knock-off, but I cannot tell whether it addresses this particular question. It would be helpful if the authors could more directly discuss (1) whether and how the bound Ca ever unbinds and (2) what might underlie the dramatic change in tail currents in the D523 mutant. Addressing these points could be done by rewriting and would not require any new experiments, (unless the authors thought of something worth testing).

2) The PKD2-L1 TRP ion channel is shown to be able to facilitate calcium influx but it is blocked by outgoing calcium ions. This unique feature of this channel is explained by a model with two calcium binding sites, illustrated on Figure 6—figure supplement 2. However, a quantitative explanation in terms of energy barriers or kinetic schemes would have enriched the paper.

3) Figure 4—figure supplement 1 shows a very important experiment and should be moved to the main text, possibly as Figure 4. In this figure, the investigators show strong evidence that outwardly-moving Ca^2+^ elicited by depolarization to above Erev for Ca^2+^ blocks the outward current. Panel B shows evidence that the outward movement of Ca^2+^ desensitizes the channel. It would strengthen the paper if the authors could show that the occurrence of desensitization also depends on the Erev of Ca^2+^ as in A.

4) In Figure 1, the authors estimate calcium levels in cells expressing PKD2-L1 transfected HEK cells and use this as evidence that calcium accumulation causes inactivation/desensitization of the channels. Whether this technique is valid was debated by the reviewers. One commented: "the relative contribution of calcium influx to the total current cannot be simply calculated based on permeability ratios. This can only be calculated, as far as I know, by simultaneously measuring calcium levels with a calcium indicator a method pioneered by Neher and used most recently by Nilius on TRPA1 channels (Karashima et al., 2010). If you look at Figure 7 from Nilius Biophys J. 2010 Mar 3;98(5):773-83, you can see that using P_Ca_ can lead to over or underestimates of fractional calcium entry by several fold in either direction, depending on the channel. Moreover, the authors do not consider the fact that the pipette serves as a large sink, reducing the rate of calcium accumulation. Thus, I think that this analysis is invalid and only weakens the paper." This reviewer added: "I would also like to see the data that supports the assertion that the currents do not desensitize when the cells are loaded with 15 mM BAPTA/5 mM EGTA." Another commented: It is okay to equate permeabilities ratios with conductance ratios. However, the authors should consider the actual driving forces since (V_m_-V_Na_) is not equal to (V_m_-V_Ca_)." The reviewers agreed that a better argument is made in Figure 1—figure supplement 1, which could be promoted to Figure 1, eliminating the current Figure 1.

5) For the experiments in Figure 2, what evidence other than single channel amplitude is there that the channel shown is indeed a PKD2-L1 channel? Does it show the same voltage dependence as the whole cell current? Is it only observed in transfected cells? Also, I am not clear what voltage was applied across the membrane (i.e. experienced by the channel) – was this -50 mV? Or +80 mV?

[Editors' note: further revisions were requested prior to acceptance, as described below.]

Thank you for resubmitting your work entitled "Atypical calcium regulation of the PKD2-L1 polycystin ion channel" for further consideration at *eLife*. Your revised article has been evaluated by Richard Aldrich as the Senior editor, a Reviewing editor, and two reviewers. The manuscript has been improved but there are some remaining issues that need to be addressed before acceptance. We are willing to offer you one more opportunity to produce an acceptable version, so we ask that you pay particular attention to the detailed comments of the reviewers below:

The revisions to the manuscript are not altogether clear, either in the writing or the scientific presentation. Please re-examine some of the changes made and edit to make the material more accessible and/or well integrated into the manuscript. Specifically, reviewers had difficulty understanding (1) the first paragraph of the Results, in part owing to grammatical/linguistic errors, as well as the new first figure, esp. regarding the voltage-dependence and its relation to other literature and the rest of the paper and (2) the response to reviewers, esp. regarding the unbinding of Ca and the two Ca-bound states in the model.

The first main issue is that the new Figure 1 shows the currents recorded in 0 divalents and presents the discovery or confirmation (it was not clear which) that this channel is voltage-dependent, even independent of divalents, which was not explicitly in the first version. As written, however, it is hard to follow, and the switch to studying divalent block is so abrupt as to be confusing. It is possible that this may be improved by clarifying some matters of nomenclature.

The second main issue is that the manuscript still shows in Figure 5 and states in the text that (with divalents) "the outward current was blocked, whereas the inward tail current remained relatively constant. These observations suggest that Ca^2+^ induced block and subsequent inactivation of all outward current, while inward flux was unimpeded." The data indeed demonstrate that repeated steps to 180 mV can make the tail current desensitize, but it is still unclear how the outward current with steps above E_Ca_ can get rid of all outward current (red traces) but still make big tails. If Ca is blocking the pore, where is the current coming from? The previous review asked about the unbinding of Ca, given the initial claim that inward current was unimpeded, and about whether the slow deactivation came from hooked tail currents with a slow rise. However, the rise times on tail currents were not addressed and neither is the profound difference in tail current deactivation times (Figure 5—figure supplement 1). Ca^2+^ binding is described as irreversible (Results, second paragraph, Discussion, third and last paragraphs), but the model of Figure 8 has a Ca-bound open-blocked state as well as a Ca-bound inactivated state, which obliquely suggests that the tails do indeed come from knock-off and that inactivation is a separate transition, although no supporting analyses were included (although they may be present in the data). The idea of two Ca bound states is not addressed anywhere in the manuscript except in the last three sentences. It was not clear to all the reviewers that two states might exist and may likewise not be clear to readers. Please edit accordingly.

---

## [Author Response]

Essential revisions:

The reviewers found the manuscript compelling. Primary questions had to do with (a) clarifying the interpretation of the mechanism, either by a more extensive Discussion accompanied by quantitative justifications and/or reanalysis to clarify the ambiguous points or by direct biophysical modeling (described in detail in points 1 and 2 below), (b) presentation of figures in a way to ensure that key data are all present in the main text (point 3 below), (c) revisiting the Ca permeability analysis (point 4 below (d) clarification of the single channel analysis (point 5 below). Please revise the manuscript accordingly.

Detailed explanation of essential revisions from reviewers' comments.

1) Results / Figure 3: "…the outward current was blocked, whereas the inward tail current remained relatively constant. These observations suggest that Ca^2+^ induced block and subsequent desensitization of all outward current, while inward flux was unimpeded." It is not clear to me how the block can be cumulative over repeated trials (steps here or ramps elsewhere) if inward current is unimpeded. This seems self-contradictory. The authors provide ample evidence that outwardly flowing Ca blocks the channel, but they do not state explicitly what happens to the bound Ca over time (upon repolarization). The data with the D523N mutant in Figure 6—figure supplement 1 shows that the lack of Ca block correlates with an extreme shortening of the long, broad tail current that characterizes the wild type. It therefore seems possible that the blocking Ca (in the wild type) is displaced by inwardly flowing ions, at least in a subset of channels, and a slow unbinding process prolongs the time course of the tail (as in displacement of quaternary ammonium ions by inward K tails in Armstrong 1971). It is hard to tell whether this idea is considered or favored or ruled out by the authors. In the Discussion there is a consideration of knock-on and knock-off, but I cannot tell whether it addresses this particular question. It would be helpful if the authors could more directly discuss (1) whether and how the bound Ca ever unbinds and (2) what might underlie the dramatic change in tail currents in the D523 mutant. Addressing these points could be done by rewriting and would not require any new experiments, (unless the authors thought of something worth testing).

We thank all reviewers for giving us the opportunity to clarify and improve this manuscript. We have addressed these questions in the last paragraphs of the Discussion section. To paraphrase, Ca^2+^ can unbind from the filter by hyperpolarizing the membrane potential but only prior to channel desensitization.

Inactivation of PKD2-L1 appears to be irreversible by membrane potential and calcium chelation. We have included a new Figure 1 and Figure 1—figure supplement 1 to address the tail current and added a scheme to Figure 8 as model.

2) The PKD2-L1 TRP ion channel is shown to be able to facilitate calcium influx but it is blocked by outgoing calcium ions. This unique feature of this channel is explained by a model with two calcium binding sites, illustrated on Figure 6—figure supplement 2. However, a quantitative explanation in terms of energy barriers or kinetic schemes would have enriched the paper.

We have added a simple kinetic scheme in Figure 8 and two paragraphs describing our model (Discussion section). Based on three kinetics quantified here we propose a four state model where the transitions between the closed to open are voltage dependent, while the inactivated state occurs after calcium-dependent block of PKD2-L1.

as in A.

*3) Figure 4—figure supplement 1 shows a very important experiment and should be moved to the main text, possibly as Figure 4. In this figure, the investigators show strong evidence that outwardly-moving* Ca^2+^
*elicited by depolarization to above Erev for* Ca^2+^
*blocks the outward current. Panel B shows evidence that the outward movement of* Ca^2+^
*desensitizes the channel. It would strengthen the paper if the authors could show that the occurrence of desensitization also depends on the Erev of* Ca^2+^

We agree and have moved these panels to the main text Figure 5. We observe desensitization when E_Ca_ is shifted very positive, like the experiments in Figure 4—figure supplement 1.

4) In Figure 1, the authors estimate calcium levels in cells expressing PKD2-L1 transfected HEK cells and use this as evidence that calcium accumulation causes inactivation/desensitization of the channels. Whether this technique is valid was debated by the reviewers. One commented: "the relative contribution of calcium influx to the total current cannot be simply calculated based on permeability ratios. This can only be calculated, as far as I know, by simultaneously measuring calcium levels with a calcium indicator a method pioneered by Neher and used most recently by Nilius on TRPA1 channels (Karashima et al., 2010). If you look at Figure 7 from Nilius Biophys J. 2010 Mar 3;98(5):773-83, you can see that using P_Ca_ can lead to over or underestimates of fractional calcium entry by several fold in either direction, depending on the channel. Moreover, the authors do not consider the fact that the pipette serves as a large sink, reducing the rate of calcium accumulation. Thus, I think that this analysis is invalid and only weakens the paper." This reviewer added, "I would also like to see the data that supports the assertion that the currents do not desensitize when the cells are loaded with 15 mM BAPTA/5 mM EGTA." Another commented: It is okay to equate permeabilities ratios with conductance ratios. However, the authors should consider the actual driving forces since (V_m_-V_Na_) is not equal to (V_m_-V_Ca_)." The reviewers agreed that a better argument is made in Figure 1—figure supplement 1, which could be promoted to Figure 1, eliminating the current Figure 1.

A fundamental and underappreciated problem inherent in voltage-clamp recordings is that the flow of ionic currents across the membrane can alter the concentrations of intracellular ions (Mathias et al. Biophysical Journal, 1990; Frankenhaeuser & Hodgkin, Journal of Physiology, 1956). When a non-selective channel with a large conductance (such as PKD2-L1) is overexpressed in a HEK cell, the internal cationic concentration is altered by accumulation of external cations during sustained activation. This effect was highlighted in a recent study examining the contribution of the pipette cationic reservoir during voltage clamp of HEK cells overexpressing P_2_X_2_ channels using a rigorous mathematical model (Li, Swartz, Nature Neuroscience 2015). The authors find that internal cation conditions are controlled initially by pipette reservoir, but after repetitive or continuous stimulation of the P_2_X_2_ channel the pipette had very little influence. The accumulation of bath internal cations (up to 100 mM) explains the observed collapse of the reversal potential, a phenomena which has also been reported in TRPV1, TRPA1, ASICs and TMEM16A channels.

While the equations in our present study to not account for ionic contributions by the pipette cation reservoir, the preceding evidence suggests that it is insignificant after continuous PKD2-L1 stimulation. A good point by the reviewer is that our study does not simultaneously measure free calcium. While having a Ca^2+^ reporter (such as a dye) may have the benefit of Ca^2+^ entry estimates, the caveat is that it introduces an additional Ca^2+^ buffer (and its Ca length constant) which complicates the free Ca^2+^ calculation.

Thus we feel the analysis in Figure 2—figure supplement 1 (previously Figure 1) is a valid approximation (not a measurement) of the free Ca^2+^ accumulation. We have adhered with the reviewer’s suggestion and substituted the Main text figure for the Figure 2 supplement (now Figure 2).

This reviewer added: "I would also like to see the data that supports the assertion that the currents do not desensitize when the cells are loaded with 15 mM BAPTA/5 mM EGTA."

This was demonstrated over a ten minute time course in Figure 2 (black traces). We have changed the text in the second paragraph of the Results section to be more clear.

5) For the experiments in Figure 2, what evidence other than single channel amplitude is there that the channel shown is indeed a PKD2-L1 channel? Does it show the same voltage dependence as the whole cell current? Is it only observed in transfected cells? Also, I am not clear what voltage was applied across the membrane (i.e. experienced by the channel) – was this -50 mV? Or +80 mV?

Good point. We have included a new figure (Figure 1—figure supplement 1)which examines the kinetics of tail currents in response to hyperpolarization and the single channel’s sensitivity to dibucaine. We have described the results as follows:

“The identity of the conductance measured in Figure 3 as PKD2-L1 is supported by the fact that such a conductance (outward 198 pS, inward 121 pS) with is not measured from untransfected HEK-293T cells and that both the single channel conductance and macroscopic whole cell currents are blocked by dibucaine (Figure 1—figure supplement 1).”

We have measured HEK currents thousands of times, have expressed almost all TRP channels in HEK cells in the past, and the properties of the observed current are only consistent with PKD2-L1. In previous work, we mutated its pore domain, knocked it down, knocked it out, and examined its temperature sensitivity, pressure sensitivity, and kinetics (DeCaen et al. 2013). The holding potential was 80 mV in Figure 1—figure supplement 1. We have removed the “≈-50 mV”- this was a typographical error.

[Editors' note: further revisions were requested prior to acceptance, as described below.]

*Thank you for resubmitting your work entitled "Atypical calcium regulation of the PKD2-L1 polycystin ion channel" for further consideration at eLife. Your revised article has been evaluated by Richard Aldrich (Senior editor), a Reviewing editor, and two reviewers. The manuscript has been improved but there are some remaining issues that need to be addressed before acceptance. We are willing to offer you one more opportunity to produce an acceptable version, so we ask that you pay particular attention to the detailed comments of the reviewers below:*

*The revisions to the manuscript are not altogether clear, either in the writing or the scientific presentation. Please re-examine some of the changes made and edit to make the material more accessible and/or well integrated into the manuscript. Specifically, reviewers had difficulty understanding (1) the first paragraph of the Results, in part owing to grammatical/linguistic errors, as well as the new first figure, esp. regarding the voltage-dependence and its relation to other literature and the rest of the paper and (2) the response to reviewers, esp. regarding the unbinding of Ca and the two Ca-bound states in the model.*

The first main issue is that the new Figure 1 shows the currents recorded in 0 divalents and presents the discovery or confirmation (it was not clear which) that this channel is voltage-dependent, even independent of divalents, which was not explicitly in the first version. As written, however, it is hard to follow, and the switch to studying divalent block is so abrupt as to be confusing. It is possible that this may be improved by clarifying some matters of nomenclature.

Previously, the reviewers suggested that we add a model describing the PKD2-L1 channel states that would benefit this manuscript. In order to do so, we needed to test the voltage dependent states of the human isoform of PKD2-L1, which has become Figure 1. Although murine PKD2-L1’s (previously called TRPP3, now designated TRPP2 by IUPHAR) voltage dependence had been partially characterized, human PKD2-L1 had not. We have clarified the nomenclature, added references and discuss the murine PKD2-L1 findings to the first paragraph for greater context. We presented these results first in order to rule out divalent-dependent effects on voltage dependent gating so that outward Ca-dependent block and subsequent inactivation could be examined in the remainder of the manuscript. We’ve substantially rewritten this section and placed subheadings to help the reader.

*The second main issue is that the manuscript still shows in Figure 5 and states in the text that (with divalents) "the outward current was blocked, whereas the inward tail current remained relatively constant. These observations suggest that* Ca^2+^
*induced block and subsequent inactivation of all outward current, while inward flux was unimpeded." The data indeed demonstrate that repeated steps to 180 mV can make the tail current desensitize, but it is still unclear how the outward current with steps above E_Ca_ can get rid of all outward current (red traces) but still make big tails. If Ca is blocking the pore, where is the current coming from?*

It is important to note that the calcium block is unidirectional. Outward PKD2-L1 current is blocked by outwardly moving Ca^2+^ ions at potentials greater than E_Ca_. This results in the loss of peak current during the depolarizing pulses (red traces) shown in Figure 5. At potentials negative to E_Ca_, Ca^2+^ does not move outwardly and peak currents are not blocked (black traces). Since Ca^2+^ blocks PKD2-L1 current in the outward direction only, the inward tail current remains comparatively large unless the slow process (τ = 1.8-2.1 s) of inactivation is initiated by multiple depolarizations positive to E_rev_ (Figure 5). We explain this unidirectional block by the relative affinities of the D523 and D525 for Ca^2+^, and hypothesize that this results from knock-on mechanisms in the inward, but not outward, direction.

The previous review asked about the unbinding of Ca, given the initial claim that inward current was unimpeded, and about whether the slow deactivation came from hooked tail currents with a slow rise. However, the rise times on tail currents were not addressed and neither is the profound difference in tail current deactivation times (Figure 5—figure supplement 1).

Thank you for clarifying your question. Currents from Kv3.1b (Labro et al. 2015) and Kv11 (as reviewed in Vandenberg et al. 2012) channels have ‘hooked tail currents’ with a slow rise time which increases from 1-10 ms over hyperpolarizing potentials. For PKD-L1, we do not see hooked tail currents. We expand the tail currents from Wt and mutant channels in Figure 6—figure supplement 1 (Previously Figure 5—figure supplement 1) to examine the rise time. From the expanding time scale, we can see that the rise time (time to peak) is within 120 μs and does not vary with membrane potential. Thus if PKD2-L1 has a fast inactivation kinetic, it is faster than we can voltage clamp the membrane. We have included a statement discussing the rise time of the tail in the second paragraph of the Results section. Note that speed of the maximum tail current is approaching the upper limit of the speed of the voltage clamp (on average 57 μs; with R_s_ = 7.1 mΩ compensated at 60% and Cm = 13.4 pF).10.7554/eLife.13413.019Author Response Image 1.**DOI:**
http://dx.doi.org/10.7554/eLife.13413.019

In Figure 6—figure supplement 1 (previously Figure 5—figure supplement 1), the speed of deactivation (decay of tail currents) is highly variable at -60 mV for all the charge carriers tested. When the data is summarized, there is no statistical difference between the time courses of deactivation for charge carriers for each mutant. The deactivation tail current kinetics are slightly faster in presence of K^+^ and Ca^2+^ compared to sodium. The exemplars are representative of the data set.

Regarding the profound differences in tail current decay: If the reviewer is referring to the differences between the Wt and filter mutant deactivation, then yes, we agree there is a 20x difference between the Wt and D523N mutant. We propose that the increased deactivation might be due to the differences in the affinity of D523 for inwardly passing ions. Asparagine is less electronegative than glutamate, and we proposed that this substitution allows cations to move more rapidly though the filter than in Wt channels – thus the channel closes faster. This implies that channel deactivation can be influenced by ion occupancy. This hypothesis is discussed in the last paragraph of the Discussion section.

Ca^2+^ binding is described as irreversible (Results, second paragraph, Discussion, third and last paragraphs), but the model of Figure 8 has a Ca-bound open-blocked state as well as a Ca-bound inactivated state, which obliquely suggests that the tails do indeed come from knock-off and that inactivation is a separate transition, although no supporting analyses were included (although they may be present in the data).

Each of the lines below discusses the process of long-term *inactivation* (irreversible on the time scale of the recordings) and occurs after I_Ca_ blockade. We cannot examine the kinetics between the blocked inactivated state and the long-term inactivated state.

“Under more physiological conditions ([Ca^2+^]_e_ = 2 mM; ~100 nM free [Ca^2+^]_i_, buffered with 5 mM BAPTA), we patch clamped PKD2-L1 expressing HEK293T cells, applied voltage ramps (-100 mV to +100 mV) and observed current potentiation followed by complete and apparently irreversible inactivation over an 8 min. time course (Figure 2—figure supplement 1 A, B).”

“Once occupied by outwardly moving calcium ions, the PKD2-L1 channels irreversibly inactivates in seconds (τ = 2.1 s).”

“Outwardly moving calcium ions rapidly block the open channel state (O_block_) and trigger a slower long-term inactivation process (K_2_). The rate of inactivation (K_2_) is dependent on outward Ca^2+^ current and is irreversible on the time scale of the experiments”.

The idea of two Ca bound states is not addressed anywhere in the manuscript except in the last three sentences. It was not clear to all the reviewers that two states might exist and may likewise not be clear to readers. Please edit accordingly.

We have emphasized this point more in the revised manuscript.